# SARS-CoV-2 Surveillance in Belgian Wastewaters

**DOI:** 10.3390/v14091950

**Published:** 2022-09-02

**Authors:** Raphael Janssens, Sven Hanoteaux, Hadrien Maloux, Sofieke Klamer, Valeska Laisnez, Bavo Verhaegen, Catherine Linard, Lies Lahousse, Peter Delputte, Matthieu Terwagne, Jonathan Marescaux, Rosalie Pype, Christian Didy, Katelijne Dierick, Koenraad Van Hoorde, Marie Lesenfants

**Affiliations:** 1Epidemiology of Infectious Diseases, Sciensano, 1050 Brussels, Belgium; 2European Program for Intervention Epidemiology Training (EPIET), European Centre for Disease Prevention and Control (ECDC), 171 83 Stockholm, Sweden; 3Infectious Diseases in Humans, Sciensano, 1050 Brussels, Belgium; 4ILEE, University of Namur, 5000 Namur, Belgium; 5NARILIS, University of Namur, 5000 Namur, Belgium; 6Department Bioanalysis, Ghent University, 9000 Ghent, Belgium; 7Laboratory for Microbiology, Parasitology and Hygiene, Antwerpen University, 2610 Wilrijk, Belgium; 8E-BIOM SA, 5000 Namur, Belgium; 9Société Publique de Gestion de l’Eau, 4800 Verviers, Belgium

**Keywords:** wastewater-based epidemiology, public health authority, surveillance, alerting indicator, SARS-CoV-2, viral load per capita, viral to faecal ratio, correlation

## Abstract

Wastewater-based surveillance was conducted by the national public health authority to monitor SARS-CoV-2 circulation in the Belgian population. Over 5 million inhabitants representing 45% of the Belgian population were monitored throughout 42 wastewater treatment plants for 15 months comprising three major virus waves. During the entire period, a high correlation was observed between the daily new COVID-19 cases and the SARS-CoV-2 concentration in wastewater corrected for rain impact and covered population size. Three alerting indicators were included in the weekly epidemiological assessment: High Circulation, Fast Increase, and Increasing Trend. These indicators were computed on normalized concentrations per individual treatment plant to allow for a comparison with a reference period as well as between analyses performed by distinct laboratories. When the indicators were not corrected for rain impact, rainy events caused an underestimation of the indicators. Despite this negative impact, the indicators permitted us to effectively monitor the evolution of the fourth virus wave and were considered complementary and valuable information to conventional epidemiological indicators in the weekly wastewater reports communicated to the National Risk Assessment Group.

## 1. Introduction

Wastewater influents collected at treatment plant inlets are a reliable parameter to monitor the health of a population as they provide markers of exposure to various chemical and biological agents such as drugs of abuse and their metabolites, viruses, bacteria, etc. [1]. Since the beginning of the severe acute respiratory syndrome coronavirus 2 (SARS-CoV-2) pandemic, many countries have started using wastewater-based epidemiology to monitor the circulation of SARS-CoV-2 [2]. The strategy owes its success to the large population represented in a single wastewater sample as well as to the high sensitivity of the analytical methods available. Wastewater enables, thus, the development of cost-effective surveillance programs, compared to individual RT-qPCR tests for monitoring the virus circulation in a large population [3].

Several studies have reported a high correlation between the SARS-CoV-2 concentrations measured in wastewater and coronavirus disease 2019 (COVID-19) cases, with 2–8-days-earlier detection by wastewater than by registration of incident cases [4,5,6,7]. The measured concentrations in wastewater were also used to estimate SARS-CoV-2 prevalence in communities [8,9,10,11] demonstrating and assessing the impact of several sources of uncertainty: shedding-related factors, population size, in-sewer factors, sampling strategy, and RNA detection [12,13]. The uncertainty related to the covered population may be addressed using the population sizes covered in the municipalities [14] or a population-normalized biomarker mass loads [15]. Another important uncertainty factor is related to the capture of rainwater in the sewage system, diluting the household viral concentrations. However, this impact can be mitigated by converting the viral concentration either in viral loads using the mean daily flow rate measured at the treatment plant inlet [14] or in viral to faecal ratios using human faecal-load indicators such as PMMoV or crAssphage [4,16,17]. Li et al. performed an uncertainty impact assessment taking into account the covered population and rainwater dilution. Their Monte Carlo simulation study concluded that using a high-frequency time-proportional sampling as well as viral concentrations corrected for rain impact, the dominant uncertainty factor is the analytical method of RNA detection in wastewater [12].

As Roka et al. concluded that higher correlation levels between the wastewater data and the number of new cases were obtained during periods of higher circulation [18], several studies stated that wastewater-based epidemiology may be a valuable tool for SARS-CoV-2 crisis management [4,19,20]. Consequently, the World Health Organization as well the European Commission have published guidance documents recommending the development of non-invasive COVID-19 surveillance programs based on wastewater analysis of the SARS-CoV-2 virus [21,22]. Up to date, almost 70 countries have implemented such systems at several scales [23]. However, according to the author’s knowledge, scientific literature lacks studies presenting wastewater-based epidemiology programs currently used to alert public health authorities. Therefore, the general objective of this work is to present the national wastewater-based epidemiology surveillance used by the public health authority to monitor the SARS-CoV-2 circulation in the Belgian population. The first specific objective is to validate the efficiency of the national wastewater surveillance by computing the correlation between the daily number of new cases and concentration in wastewater corrected for rain impact and covered population size. The second one is to assess the potential of the wastewater indicators to detect resurgences and peaks of COVID-19 circulation.

## 2. Materials and Methods

### 2.1. Sample Collection, Concentration, Extraction, and Quantification

Samples were collected at the inlet of 42 wastewater treatment plants located in Belgium. The plants were selected aiming to cover areas with high population densities, including plants covering a large population, and selecting a minimum of two plants per province. In total, the wastewater surveillance covers 5 million inhabitants, or 45% of the Belgian population. The number of inhabitants connected to the plants was provided by the regional wastewater agencies.

Between the 15 September 2020 and 1 December 2022, 4984 SARS-CoV-2 samples were collected in total. The sampling frequency was designed to monitor the situation twice a week as recommended in the European Commission recommendation [22]. Sampling days were selected when the population’s mobility was high, which is during weekdays. Hence, samples were collected twice a week, on Mondays and Wednesdays, using 24 h time-proportional auto-samplers. Samples were collected simultaneously in the plants ensuring temporal comparability. Samples were stored at 4 °C and transported to one of the three laboratories for analysis within 24 h. Aliquots of wastewater samples were frozen at −20 °C for any requirements of retrospective analyses. The population covered by the treatment plants, and the corresponding laboratory, can be seen in Appendix A, as well as a map showing the localization of the plants in Appendix A.

For the sample preparation, the laboratories of Sciensano and UAntwerp used the method I, while the laboratory of E-BIOM used the method II. The method I was based on a preliminary study conducted by Boogaerts et al. to compare the performance of different sample preparation methods [24]. Briefly, wastewater samples of 50 mL were centrifuged (4654× *g*, 30 min, 4 °C). The supernatant was concentrated by ultrafiltration Centricon Plus-70 Centrifugal filters (100 kDa and 30 kDa; 1500× *g*, 35 min, 4 °C) and immediately proceeded to viral RNA extraction using the Maxwell^®^ RSC PureFood GMO and Authentication Kit (Promega, Madison, WI, USA). The method II was based on a method described by Coupeau et al. [25]. Briefly, wastewater samples of 100 mL were centrifuged (2500× *g*, 25 min, 4 °C). The 60 mL of the supernatant was concentrated by ultrafiltration (Amicon Ultra-15, 10 kDa; 3200× *g*, 35 min) and immediately proceeded to viral RNA extraction using a Trizol extraction method.

For the quantification of three SARS-CoV-2 gene fragments, RT-qPCR was used in the present study targeting the nucleocapsid (N1, N2), and the virus envelope (E). The eluted RNA was used for one-step RT-qPCR with TaqMan™ Fast Virus 1-Step Master Mix (Thermofisher, Waltham, MA, USA ) in the method I and with Takyon™ One-Step No Rox Probe 5X MasterMix dTTP in method II. The following conditions were used: one cycle at 50 °C for 5 minutes (RT) and one cycle at 95 °C for 20 s (Taq polymerase activation); 45 cycles at 95 °C for 5 s (denaturation) and 60 °C for 30 s (annealing/elongation).

Quantification of PMMoV was used as a faecal indicator. The eluted RNA was used for one-step RT-qPCR with TaqMan™ Fast Virus 1-Step Master Mix (Thermofisher, Waltham, MA, USA). The amplification was performed under the following conditions: one cycle at 50 °C for 5 minutes (RT) and one cycle at 95 °C for 30 s (Taq polymerase activation); 45 cycles at 95 °C for 5 s (denaturation) and 56 °C for 60 s (annealing/elongation). Amplification and quantification were performed using stepOne plus (applied biosystems), AriaMx (Agilent, Santa Clara, CA, USA), and Lightcycler 480 II (Roche, Basel, Switzerland) qPCR thermocyclers.

All assays were performed at least in duplicate. Four different sets of primers and probes were used as described in Table 1. A standard curve of a 5-fold serial dilution of SARS-CoV-2 RNA (NIBSC code 19/304) and PMMoV gBlocks™ Gene Fragment (IDT, Coralville, IA, USA) was used.

The limit of quantification (LOQ) was assessed using the NIBSC 19/304 reference material and 20 replicates. The calculation was performed with the R software (version 4.1.0) [29] using the script provided by Merkes et al. [30]. The resulting LOQ varied between 8 and 16 copies/mL of wastewater for the three laboratories. A conservative estimated LOQ was fixed at 20 gene copies/mL.

The performance of the three laboratories was assessed by an inter-laboratory trial using naturally contaminated wastewater samples. The intra-laboratory results were very reproducible, while the inter-laboratory results showed some larger variations, especially for the higher contaminated samples. The viral concentrations obtained by different laboratories were not compared but normalized based on a reference period. The normalization methodology, which is an important part of the present work, is described in Section 2.2.

### 2.2. Concentrations in Wastewater and Indicators

The methods to obtain the concentrations in wastewater improved throughout the surveillance. Between 15 September 2020 and 15 February 2021, the analytical methods provided non-quantitative results expressed as Ct values. Therefore, the measured Ct values obtained by the laboratories during that period were transformed into estimated viral concentration (SARS-CoV-2 RNA gene copies/mL) retrospectively using the mean parameters of ten calibration curves provided by the corresponding laboratory. The mean values and standard deviation of the calibration curves obtained at the start of the quantification period were stable. From 15 February 2021 onwards, the analytical methods were adapted to quantify viral concentrations (SARS-CoV-2 RNA gene copies/mL).

The mean of the N1, N2, and E gene fragments was used as the quantified viral concentration and the mean of the PMMoV quantification was used as the faecal indicator concentration. The faecal indicator is known to undergo no substantial seasonal fluctuation [31]. The latter was used to account for the variation in inhabitants covered by the treatment plants and the wastewater dilution. Spearman correlation coefficients computed for the three combinations of quantified viral concentration of targeted SARS-CoV-2 gene fragments (N1 vs. N2, N1 vs. E, N2 vs. E) are represented in Appendix A grouped by month for the quantified period. High levels of correlations can be observed in Appendix A for all combinations. All correlations were significant (*p* < 0.0001). The lowest correlation coefficient, being 0.83, was obtained for the N2 vs. E combination in June 2021 when the viral concentrations were close to the LOQ. It was concluded from this correlation study that the information conveyed by a gene fragment is similar to the others.

According to the work of Bertels et al. [13], several factors can affect the SARS-CoV-2 concentrations measured in wastewater, such as rainwaters captured in the sewage system or the variation in the population size connected to the treatment plants. Therefore, in the present work, two different methodologies were used to account for the rain and the population size impacts. The two methodologies were applied when quantified viral concentrations were available. Firstly, the quantified viral concentrations were expressed as viral load per capita (SARS-CoV-2 RNA copies/day/100 k inhab.) by multiplying the quantified viral concentration (SARS-CoV-2 RNA gene copies/mL) by the mean daily inflow rate (m^3^/day) provided by regional wastewater agencies and normalized with the number of inhabitants connected to the corresponding treatment plant. Secondly, the quantified viral concentrations were expressed as viral to faecal ratio (SARS-CoV-2 RNA gene copies/PMMoV RNA gene copies) by dividing the quantified SARS-CoV-2 concentration (SARS-CoV-2 RNA gene copies/mL) by the faecal indicator concentration (PMMoV RNA gene copies/mL). As both concentrations are measured in the same sample, computing their ratio cancels the dilution impact caused by rain events.

The epidemiological situation in the 42 areas covered by the wastewater surveillance is assessed twice a week thanks to three indicators computed for each individual area: High Circulation, Fast Increase, and Increasing Trend. The three indicators can be computed based on the quantified viral concentration, the viral load, or the viral to faecal ratio. Here is a description of the indicators in case they are computed on the quantified viral concentrations. The High Circulation indicator provides information on the level of SARS-CoV-2 concentration, the Fast Increase indicator highlights the areas where the viral concentrations are increasing quickly in a short term, and the areas where the concentrations rise for a longer term are indicated by the Increasing Trend indicator.

In further detail, before computing the indicators, a normalization step is applied to the viral concentrations. The normalization allows for a comparison with a reference period as well as between analyses performed by distinct laboratories. The normalization is performed by expressing the viral concentration in wastewater in percentages of the maximum value recorded during a previous virus wave for each area. It is worth mentioning that the maximum value recorded in a specific area depends on the epidemiological context, and thus varies according to the reference period considered.

The format of the indicators is Boolean. Therefore, if the normalized viral concentration exceeds half (50%) of the highest value recorded during a previous virus wave, the value of the High Circulation indicator is set to 1. Else it is set to 0. The value of the second indicator, called the Fast Increase, is set to 1 if the moving average on the past 7 days of the normalized concentration in wastewater has increased more than 70% over the past week. Else it is set to 0. Additionally, the value of the third indicator, called the Increasing Trend, is set to 1 if the moving average on the past 14 days of the normalized concentration in wastewater has increased for more than 6 days. Otherwise, it is set to 0. The graphical abstract shows a geographical representation of the indicators using pie chart pictograms.

The indicators computed based on the viral concentrations can be significantly underestimated in case of rain events as rainwaters captured in the sewage system can dilute the viral concentrations. This is not the case for the indicators computed on the viral loads and viral to faecal ratios, as these signals correct for rain dilution. Therefore, the impact of rain events on the indicators was assessed by comparing the indicators computed on the viral concentrations with the ones computed on the viral loads and viral to faecal ratios. If the sum of areas positive to the indicators computed on the viral concentrations was lower than the sum computed on the two other signals, then the sample was estimated to be impacted by a rain event.

Up to date, the wastewater indicators are computed on the viral concentrations because logistical and technical issues do not permit the collection on time of inlet flow rates and faecal indicator concentrations ensuring a weekly publication. Therefore, the viral concentrations measured in the Monday and Wednesday wastewater samples are updated within 48 h after sampling, twice a week, on the Belgian public health dashboard (link provided in reference) [32]. Once a week on Tuesday, a wastewater-based epidemiology report is produced discussing the situation observed at several geographical scales. The report is published on the Belgian public health website serving the national risk assessment group in evaluating the epidemiological situation (link provided in reference) [33].

The weighted mean viral concentrations, the viral load, or the viral to faecal ratios (*x_µ_*) are computed at the national, regional, and provincial scales thanks to Equation (1), using the population size (*Pop_i_*) covered by each treatment plant (*i*), and the viral concentrations, the viral load, or the viral to faecal ratios (*x_i_*) measured in the corresponding treatment plant.
(1)$$\begin{array}{c} \mathrm{sdaynt}\;\mathrm{planthe}\;\mathrm{corresponding}\;\mathrm{sample}\;\mathrm{viral}\;\mathrm{load},\;\mathrm{or}\;\mathrm{viral}\;\mathrm{to}\;\mathrm{faecal}\;\mathrm{ratio}\;\mathrm{is}\;\mathrm{computed}\;at\;the\;national,\;regionalx_{µ}\\ =\underset{i}{\sum }\left(x_{i}*IE_{i}\right)/\underset{i}{\sum }\left(Pop_{i}\right)\; \end{array}$$

### 2.3. COVID-19 Cases in the Covered Areas

The COVID-19 case data were obtained through the national SARS-CoV-2 surveillance system set up by the Belgian public health institute (Sciensano, Brussels, Belgium) [34]. Cases recorded in the catchment area of a particular treatment plant ($case_{i}$) were computed as follows because a treatment plant can cover only partially a municipality:(2)$$case_{i}=\underset{i}{\sum }\frac{inhab_{muni\_covered,\;i}}{inhab_{muni\_tot,\;\;i}}\;case_{muni\_tot,\;\;i}\;$$
with the different municipalities covered by a plant (i), the total number of inhabitants in the municipality ($inhab_{muni\_tot,\;i}$), the number of inhabitants covered by the plant in the municipality ($inhab_{muni\;covered,\;i}$), and the total number of COVID-19 cases recorded in the municipality ($case_{muni\_tot,\;i}$). This methodology was applied to the 42 treatment plants included in the surveillance.

### 2.4. Data Analysis

Data management was performed using SAS (SAS 7.15, NC, USA) while statistical analysis and visualization were carried out with the R software (version 4.1.0) [29]. The concentrations in wastewater had a non-normal distribution. Spearman correlation coefficients were computed between the SARS-CoV-2 concentration measured twice a week in wastewater samples and the daily new COVID-19 cases registered on the same days. The correlations were computed on non-averaged raw data. For correlation computation, data of the 42 treatment plants were used and were, thus, not aggregated at a national level. Additionally, the periods of COVID-19 waves were selected visually from Figure 1.

## 3. Results

### 3.1. Concentration in Wastewater and Daily New Cases

The estimated viral concentrations, quantified viral concentrations, viral loads, and viral to faecal ratios measured in wastewater samples are represented in Figure 1 together with daily new cases aggregated for all areas covered by the surveillance. The peak observed in November 2020 in the wastewater through the estimated viral concentrations corresponds to the second SARS-CoV-2 wave that occurred in Belgium. This second wave was followed by a third and a fourth wave in March 2021 and December 2021, respectively. The third wave led to lower daily new cases than the fourth wave. Between the third and fourth waves, a smaller peak was observed between mid-August and mid-September 2021.

The high levels of correlation observed in Figure 1 were computed statistically in Table 2, all being significant (*p* < 0.0001). Results show that higher correlation coefficients were obtained during the occurrence of a wave. Additionally, when available, the viral load per capita demonstrated lower correlation coefficients than the viral concentration and viral to faecal ratio. Results of the wastewater indicators, used to monitor the SARS-CoV-2 circulation, are presented in the next section.

### 3.2. Wastewater Indicators

The epidemiological evolution of the fourth wave was assessed based on the three indicators computed based on the normalized viral concentrations with the third wave as the reference period (15/02/2021–01/04/2021). Normalization of the concentrations was performed to allow for a comparison between the concentrations measured during the previous wave, as well as between analyses performed by distinct laboratories. The number of catchment areas sampled and positive to the different indicators over time is represented in Figure 2.

The evolution of the indicators during a wave is typically the following: (i) at the start of the resurgence, the number of areas with the Fast Increase and Increasing Trend indicators positive rose quickly; (ii) then, the increasing concentration in several areas led to the fulfilment of the High Circulation indicator’s condition; (iii) at the peak, the Fast Increase indicator number started to decrease, followed the next week by a decrease in the number of areas in which the Increasing Trend indicator was positive. This pattern can be observed in Figure 2 for the third wave period. Afterward, a decrease in the indicators up to their lowest levels was seen, between April and the end of June 2021, corresponding to the third wave’s depletion. From July to October 2021, the number of areas for which the Increasing Trend indicator was positive remained stable. At the beginning of October 2021, a significant surge in the Fast Increase and Increasing Trend indicators indicated the beginning of the fourth wave. Then, in mid-November 2021, the fourth wave’s peak was detected by a drop in the Fast Increase indicator followed the next week by a reduction in the Increasing Trend indicator.

The Risk Assessment Group, led by the National Public Health Institute and composed of experts with different backgrounds, was informed of the evolution of the situation observed in the wastewater. The alerts communicated to the assessment group were contextualized with the sources of uncertainty impacting the wastewater indicators.

To assess if some sample dates were impacted by rain events, the sum of areas positive to the indicators computed on the viral concentrations (Figure 2) was compared with the indicators computed on signals accounting for the rain dilution: the viral loads (Appendix A) and viral to faecal ratio (Appendix A). This exercise concludes that several sampling dates were impacted by rain events causing an underestimation of the wastewater indicators computed on viral concentrations. The dates impacted by rain events are marked with blue bars in Figure 2.

## 4. Discussion

### 4.1. Concentration in Wastewater and Daily New Cases

The second, third, and fourth waves can be seen in wastewaters in Figure 1 together with a smaller rise in concentration occurring between mid-August and mid-September 2021. This small increase observed at the national level was caused by a significant increase in the concentration in the provinces of Brussels and Liège. Therefore, this event was localized and did not spread to the entire country.

The measurement uncertainty related to the viral concentrations was reduced when the quantification of the viral concentrations began on 15 February 2021. On the same date, the computation of the viral loads and the viral to faecal ratios was started. The differences observed in Figure 1 between the viral concentrations, viral loads, and viral ratios can be explained by the variation in flow rates and PMMoV concentrations. The variations in flow rates are mainly caused by rain events, while the variations in PMMoV concentration are induced by rain events as well as the mobility of people excreting PMMoV in the sewage system.

Despite the uncertainty caused by the estimation of viral concentrations between 15 September 2021 and 15 February 2022, a correlated rise was observed in the concentrations in wastewater and the daily new cases during this period, illustrating the second wave. A simultaneous rise was also observed during the third and fourth waves. However, the daily new cases recorded at the peak of the third wave were expected to be higher, given the high viral concentrations measured at that time. The height of the peaks recorded in wastewater and daily new cases was influenced by several parameters. Firstly, from April to December 2021, the proportion of Delta variant circulating rose from 0% to 99%. The Delta variant is associated with increased faecal loads and longer shedding duration than the Alpha variant [35]. During the same period, the proportion of the Belgian population fully vaccinated ramped from 4% to 75% [33], and vaccination is known to reduce viral loads when having breakthrough infections [36]. Mobility restriction measures, used to control infections, were more stringent in April than in December 2021 [33]. Additionally, the motivation for adherence to restriction measures evolved, being moderate in April and high in December 2021 [37]. Another important parameter influencing the daily new cases recorded is the modification of the testing strategy. However, this one remained constant between April and December 2021 [33]. These combined effects may explain similar viral concentrations measured during the peaks of the third and fourth wave, although fewer daily new cases were registered at the peak of the third wave as compared to the fourth one.

Several studies have reported high levels of correlation between concentrations in wastewater and COVID-19 cases [4,5,6], similarly to the results obtained in Table 2 in the entire period. This relation is caused by the presence of virus fragments in the stool of cases infected by SARS-CoV-2 [38]. During waves, Roka et al. have reported higher correlation coefficients than in low-circulation times [18]. The present work confirmed the finding of Roka et al. as all the correlation coefficients computed on viral concentrations during the wave episodes were higher than the ones for the entire period. Additionally, the highest correlation coefficient was obtained during the second wave (0.66), compared to 0.59 and 0.57 during the third and fourth waves, respectively. Additionally, several studies have reported taking into account a lag time, because of the incubation period and excretion period [4,5,6,7].

Correlations can be computed not only on the viral concentrations but also on the viral loads per capita and the viral to faecal ratios. However, expressing the viral concentrations into viral load or viral ratio did not significantly improve the correlation coefficients at a national level for the different periods considered. Further investigations will be dedicated to the understanding of this impact at a smaller spatial scale (e.g., areas covered by a treatment plant). This is important as the correlation coefficients may strongly depend on the local context such as the testing strategy in place, the proportion of rainwater captured in the sewage system, the proportion of industrial wastewater or other in-sewer factors, the architecture of the sewage system or/and the mobility of the population covered. For instance, mobility could be an important influencing factor impacting the correlation results in touristic places during holiday periods, university campuses, industrial zonings, or cities with international connections. Additionally, mobility levels significantly increased at the reopening of national and international travel. Therefore, future work will be devoted to understanding the preceding character of the viral load or viral ratio for the 42 covered areas. This exercise will be performed using an autoregressive integrated moving average (ARIMA) model including mobility factors, vaccination rate, variant circulation, and other parameters influencing the prediction of daily new cases.

### 4.2. Wastewater Indicators

The epidemiological evolution of the fourth wave was assessed based on the three indicators computed with the third wave as the reference period. As presented in Figure 2, throughout the summer holidays (July–August 2021), virus circulation steadily increased from a low level. A simultaneous surge in the indicators in wastewater and the daily new cases was observed at the beginning of October, corresponding to the start of the fourth wave. Additionally, the peak of the fourth wave was observed at a similar moment throughout the wastewater surveillance and the case-based surveillance. Therefore, the wastewater indicators were assessed to be an effective tool by the national public health authorities for monitoring the virus circulation. Future work will be devoted to the improvement in the wave’s detection via the indicator’s evolution at a local scale.

At the fourth wave’s start, three consecutive samples were diluted by the rain collected in the covered sewage systems, as the number of areas for which the three indicators were fulfilled was underestimated when compared to the indicators correcting for the rain dilution. The wastewater surveillance’s ability to alert early the authority was, thus, impacted by the rain and an additional week was required to confirm the alerting trend. The delay caused by the rain did not prevent the wastewater surveillance from providing valuable information which supported the epidemiological situation observed through the case surveillance. Throughout the wastewater surveillance, several advantages over case-based surveillance were highlighted. The analytical methods used in wastewater surveillance do not change over time and are not limited by a maximum capacity, as is the case for the testing strategy. However, the wastewater surveillance does not allow for a case-based approach and follow-up of COVID-19.

According to the review by Bertels et al. [13], several factors can influence SARS-CoV-2 concentrations in wastewaters: shedding-related factors, population size, in-sewer factor, sampling strategy, and analysis. Shedding-related factors include the variation in virus shedding mass and rate. The population size covered by the treatment plants may differ daily due to commuting activities. Amongst in-sewer factors stand the load and properties of solid particles and organic matter, water properties, as well as the influx of rainwater. Sampling strategy and analysis-related factors include sampling frequency, sampling mode, sample transportation, and laboratory analytical methods.

Rainwaters entering the sewage system are widely reported to cause the dilution of viral concentrations [4,14,16,17]. Similar results were found in the present work as several sampling dates were impacted by rain events, causing an underestimation of the wastewater indicators computed on viral concentrations. Besides the in-sewer factors, the impacts related to the population sizes, sampling strategy, and analysis were estimated to be strongly reduced by the fact that the indicators were computed on normalized concentration. As the population size, the sampling strategy, and the analytical method associated with a catchment area remained constant throughout the wastewater surveillance duration, the maximal viral concentration used for the normalization step eliminated the corresponding uncertainty factors. To conclude, the rainwater entering the sewer system was assessed to be the major source of uncertainty impacting the wastewater indicators computed on the viral concentrations. To date, the wastewater indicators are computed on the viral concentrations because logistical and technical issues do not permit the collection on time of the inlet flow rates and faecal indicator concentrations for a weekly publication. However, actions are taken to be able, in a near future, to report the indicators based on the viral load or viral to faecal ratio.

To conclude, the high correlations obtained in the present study laid the first stones for the development of the first Belgian national wastewater-based epidemiological surveillance program. The underestimation of the wastewater indicators did not prevent the wastewater surveillance program to report valuable and complementary information to conventional epidemiological indicators. The indicators computed on the viral concentration permitted the monitoring of the fourth wave’s evolution, thanks to the important role of the normalization step. Therefore, the wastewater indicators were assessed to be an effective tool by the national public health authorities for monitoring SARS-CoV-2 circulation in the Belgian population. Future work is envisioned to quantify the possible preceding character of the concentrations in wastewater at a local scale, as well as to reduce the impact of rainwater on the wastewater indicators. Additionally, the potential of wastewater-based epidemiology will be disseminated to a broader audience through the integration of the wastewater indicators in the national public health dashboard.

## Figures and Tables

**Figure 1 viruses-14-01950-f001:**
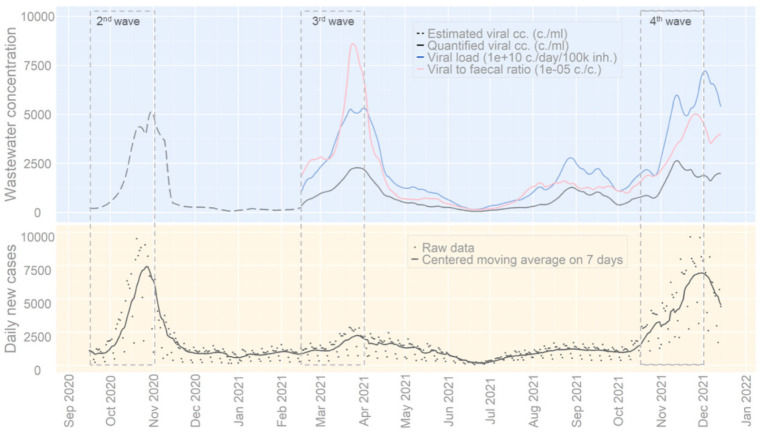
SARS-CoV-2 concentrations in wastewater and COVID-19 cases aggregated for all areas covered by the Belgian wastewater surveillance over time. The estimated and quantified viral concentration, the viral load per capita, and the viral to faecal ratio are centred moving average on 14 days. The raw data and centred moving average on 7 days of the daily new cases are presented in the lower graph.

**Figure 2 viruses-14-01950-f002:**
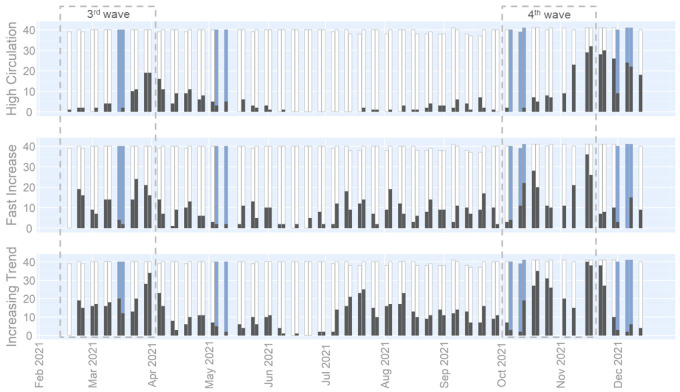
Evolution of the number of areas for which the wastewater indicators are positives. The indicators are computed on the viral concentration (SARS-CoV-2 RNA gene copies/mL) with the 3rd wave as reference period (15/02/2021–01/04/2021): High Circulation, Fast Increase, and Increasing Trend. The number of areas for which each indicator is positive or not is indicated by black and white bars, respectively. Wastewater samples impacted by the rain are marked with blue bars. Total number may be lower than 42 when technical issues prevented samples to be taken for some of the wastewater treatment plants.

**Table 1 viruses-14-01950-t001:** Overview of primers/probes sequences used for the RT-qPCR assays.

Gene	Primer/Probe	Final cc.	Sequence (5′-3′)	Ref.
N1	nCoV_N1-F	500 nM	GACCCCAAAATCAGCGAAAT	[26]
	nCoV_N1-R	500 nM	TCTGGTTACTGCCAGTTGAATCTG	[26]
	nCoV_N1-P	125 nM	ACCCCGCATTACGTTTGGTGGACC	[26]
N2	nCoV_N2-F	500 nM	TTACAAACATTGGCCGCAAA	[26]
	nCoV_N2-R	500 nM	GCGCGACATTCCGAAGAA	[26]
	nCoV_N2-P	125 nM	ACAATTTGCCCCCAGCGCTTCAG	[26]
E	E_Sarbeco-F	400 nM	ACAGGTACGTTAATAGTTAATAGCGT	[27]
	E_Sarbeco-R	400 nM	ATATTGCAGCAGTACGCACACA	[27]
	E_Sarbeco-P	200 nM	ACACTAGCCATCCTTACTGCGCTTCG	[27]
PMMoV	PMMV-rev-F	400 nM	GAGTGGTTTGACCTTAACGTTTGA	[28]
	PMMV-R	400 nM	TTGTCGGTTGCAATGCAAGT	[28]
	PMMV-P	200 nM	CCTACCGAAGCAAATG	[28]

**Table 2 viruses-14-01950-t002:** Spearman correlation coefficients between the daily new COVID-19 cases and SARS-CoV-2 concentration in wastewater: viral concentrations, viral loads per capita, and viral to faecal ratios. Different periods were selected: the entire period (15/09/2020–15/12/2021), 2nd wave (15/09/2020–01/11/2020), 3rd wave (15/02/2021–01/04/2021), and 4th wave (15/10/2021–01/12/2021). The number (N) of data included in the analysis is shown. All correlations were significant (*p* < 0.0001).

Daily New Cases Correlation Against	Entire Period (N = 4984)	2nd Wave (N = 507)	3rd Wave (N = 558)	4th Wave (N = 492)
Viral concentration (SARS-CoV-2 gene copies/mL)	0.54 ^1^	0.66 ^1^	0.59	0.57
Viral load per capita (SARS-CoV-2 gene copies/day/100 k inhab.)	n.d. ^2^	n.d. ^2^	0.52	0.52
Viral to faecal ratio (SRAS-CoV-2 gene copies/PMMoV gene copies)	n.d. ^2^	n.d. ^2^	0.55	0.58

^1^ Estimated viral concentration data were used from 15/09/2020 to 15/02/2021 when quantified viral concentrations were not available. ^2^ n.d. denotes not determined. Viral loads per capita and viral to faecal ratios were computed when quantified viral concentrations were available.

## Data Availability

Not applicable.

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
