# Peer review of "SARS-CoV-2 Surveillance in Belgian Wastewaters"

_viruses, 2022, doi:10.3390/v14091950_

Round 1
Reviewer 1 Report (Previous Reviewer 2)
I accept the responses to the comments and the revised version of the manuscript.
Author Response
Dear reviewer 1, we are pleased to read that you accept the response to the comments and revised version. However, the revised version had to be modified due to comments received from other reviewers. The modifications done are aligned with the comment you provided us earlier.
Reviewer 2 Report (Previous Reviewer 3)
Authors made efforts to thoroughly address reviewers comments.
Author Response
Dear reviewer 2, we are pleased to read that our efforts to address your comments were accepted. However, the revised version had to be modified due to comments received from other reviewers. The modifications done are aligned with the comment you provided us earlier.
Reviewer 3 Report (Previous Reviewer 1)
General comments:
The objective of this study was to describe the implementation of an effective national wastewater-based epidemiological surveillance system to assist the public health authority in monitoring the circulation of SARS-CoV-2 in the Belgian population. On the one hand, the correlations between the daily number of new cases and the rain-corrected SARS-CoV-2 concentration in wastewater and the size of the population covered are explored. On the other hand, three different wastewater indicators are proposed in relation to the national context of SARS-CoV-2 circulation. Epidemiological data of SARS-CoV-2 based on wastewater are already widely described in the literature, but this paper proposes new alternative indicators, namely high circulation, rapid increase and increasing trend, for the assessment of the health situation and the prediction of resurgence and peak circulation of the virus.
The article is very descriptive and does not really raise any scientific questions. The article is well presented and organized but suffers from impressions that sometimes make the reading of the article difficult and not fluid. Further detail should be given on different sections for clarification, and English proofreading should be made throughout the manuscript.
Specific comments:
Line 73: “actual water usage data”. Please rephrase, not clear at all.
Line 95: recommendations provided by the EU commission should be also mentioned here.
Line 110: I do not understand this point. What do you mean by “time-scale of COVID-19 outbreaks of several week”? Moreover, the sampling frequency is fixed: twice a week. Please rephrase or clarify.
Lines 140-146: Why were different cut-off (100, 30 and 10 kDa) used for the ultrafiltration devices used in the different laboratories? Have you assessed the impact of these differences on the recovery rate of SARS-CoV-2? What are the recovery rates for each laboratory?
Lines 153-154: Are the RT-qPCR amplification conditions the same for both MasterMix?
Line 157: Which Mastermix is used for PMMoV?
Line 161: thermocyclers, with a “s”.
Line 166: please modify the table caption. Replace analytical methods by RT-qPCR assays
Line 176: Reference for the R software?
Line 177: Please precise the LOQ for each individual RT-qPCR assay, namely N1, N2 and E gene target, for each laboratory. RT-qPCR efficiencies will be also welcome.
Line 180: “remained high”. This does not mean anything. Please add the value to convince the reader. Data not shown is inappropriate her.
Lines 186-187: please provide values on the intra- and inter-laboratories trial.
Line 189 and in the whole document (line 190, line201…): Concentration of what? Please replace “concentration in wastewater” by “SARS-CoV-2 concentration in wastewater”
Line 209: provide the units of the SARS-CoV-2 and PMMoV concentration.
Lines 220-221:” In this work, concentration in wastewater is used as a generic term referring to viral concentration, viral load per capita, and viral to faecal ratio”. This does not seem appropriate at all and is very confusing for the further reading of the results. It is never clear to which values, data (virus concentration, viral load or ratio) the results refer. Furthermore, whether in figure 1 or figure 2 and the appendices, the data are presented separately. The text must be modify to clarify the reading and the understanding.
Lines 222-224: Normalization remains very unclear to me. I do not understand its real interest
Lines 228-235: What are the units (scale) of the three indicators?
Lines 236-237: Why this sentence is here?
Line 245: Which concentration (viral concentration, viral load or viral ratio)? It’s very confusing here, and very important to know to understand the equation 1. Not sure that the equation is applicable of all the three types of data.
Line 246: What represent the IE value?
Line 283: please clarify this sentence: level of what, which waves????
Table 2: Why there is a 1 on the correlation of the viral concentration for the entire period? Is it the whole period or not?
Line 295: “during between” ???, 15/09/2020?
Line 298: indicators, with a “s”
Line 299: specify sampling areas.
Line 301: why the 3rd wave was selected as the reference period. Please explain
Line 336-337: Please clarify your conclusion. Why?
Figure 2 (+B1 and B2): please modify the legend of the y-axis. It is the number of the sampling area. Not the indicators. Specify the unit of the viral concentration.
Line 348-349: Rephrase. No sense
Lines 426-428: Why? Please explain
Line 439: First Belgian national WBE surveillance. Specify the country, since similar national surveillance tool are available for numerous other countries.
Author Response
Please see the attachment.

Reviewer 4 Report (New Reviewer)
This paper presents a study related to Wastewater-based surveillance to monitor the SARS-CoV-2 circulation in the Belgian population. The study is based on three alerting indicators included in the weekly epidemiological assessment. Authors investigated the High Circulation, Fast Increase, and Increasing Trend.
Foremost, I suggest removing all internal comments between authors for a manuscript before submission. The manuscript reports several comments and corrections that are not useful for reader, or reviewer.
The introduction mentioned several studies related to SARS-CoV-2 (COVID-19) cases. However, it needs to be completely rewritten to explain the general objectives, synthesize the related work and its limitations, deduce thereof the specific objectives, and summarize how they have been addressed. In the current version, the introduction contains a number of basic statements useless in a high-level publication, while large pieces of information are missing. Furthermore, I suggest reporting also some line of introduction related to COVID-19, for instance authors could cite this study related to epidemiology of Covid-19 in several countries: doi.org/10.3390/e24070929 I suggest including it within the references, in that it highlights the relevance of the policies adopted by countries in the last years, as well as the trend related to positive cases and several useful information to present SARS-CoV-2 (COVID-19).
For instance, the lines between 92-106 (and many others) are misleading. I considered these as a part of the introduction, however I think that these are related to materials and methods. I suggest cleaning the manuscript. Furthermore, I suggest also checking typos and broken sentences, as well as to improve the presentation: extensive editing of English language and style is required.
“Materials and Methods” presents a lot of information that should be thorough. This section is the most frustrating one. It should provide a sufficiently precise and rigorous description of the proposed method, so that interested readers could reproduce it. To give an example, lines between 150 and 165 are very misleading; authors mentioned “ultrafiltration (Centricon Plus-70 Centrifugal filters (100 kDa and 30 kDa; 1 500 x g, 35 min, 4 °C)”, “TakyonTM One-Step No Rox Probe 5X MasterMix dTTP”, “Maxwell® RSC PureFood GMO”, “The eluted RNA was used for one-step RT-qPCR with TaqManTM Fast Virus 1-Step Master Mix (Thermofisher) in method I and with TakyonTM One-Step No Rox Probe 5X MasterMix dTTP in method II”… why? Parameters, methods, and methodology seem arbitrary. What are the criteria that led to these choices? Why these and not others? Are there other studies on this subject that present similar parameters? If the parameters vary, will the result improve or worsen? In my opinion, the justification of various methodology implementation choices should be provided.
Results need of a statistical analysis to demonstrate its validity. Statistical significance is missing. It does not demonstrate (statistically) whether the outcome of the processing is reliable enough.
The text needs to be thoroughly revised for numerous typos and grammatical mistakes.
In my opinion, the manuscript is not ripe for acceptance and should be rejected. Nonetheless, the study appears to have some basis, and I would propose to reevaluate it after a major ("major-major") revision.
Round 2
Reviewer 4 Report (New Reviewer)
The authors have completed several revisions, based on most of the proposed comments / suggestions. I suggest including statistical significance within the table/s of interest, for each data.
The paper is exhaustive. However, the minors are discretionary, even if particularly recommended.
Minors:
- statistical significance within the table/s of interest, for each data
- double spell checking and typos.
This manuscript is a resubmission of an earlier submission. The following is a list of the peer review reports and author responses from that submission.
Round 1
Reviewer 1 Report
General comments:
The objective of this study was to describe the national wastewater-based surveillance used by the public health authority to monitor the circulation of SARS-CoV-2 in the Belgian population. First, the correlations between the daily number of new cases and the rain-corrected sewage concentration and the size of the population covered are explored. Secondly, sewage indicators are developed in relation to the national context of SARS-CoV-2 circulation. Although sewage-based epidemiological data for SARS-CoV-2 are already widely described in the literature, this paper has the merit of attempting to provide new alternative indicators, namely high circulation, rapid increase and increasing trend, for health situation assessment.
In general, I think that the article suffers from many inaccuracies and methodological gaps, which do not facilitate the understanding of the results and make it difficult to follow the reading of the article. The materiel and method section should be rewritten with many additional information useful for the understanding of the results (see details below). This aspect needs to be greatly improved before a possible publication. In additional, the dataset and the huge quantity of quantitative data of the study was not highlighted at all. On the other hand, the data collected are quite complex due to numerous different extraction and analytical methods, and different laboratories involved. How all these are comparable? To what extent they can be pool and analyzed altogether? All these aspects should be mentioned and discussed in the paper.
The interest of the study is not really questioned, as it deals with a topical subject, but the study remains very descriptive, without any real scientific objective and research hypothesis. This aspect should also be reworked and improved to bring out a real objective and new interest. One wonders what this study brings in comparison to the many others already published. This is not highlighted enough.
Specific comments:
Line 43: not wastewater effluents, but wastewater influents. The samples were collected at the inlet of the WWTP.
Line 51: individual RT-PCR, not PCR
Line 70: “actual water usage data”. What does it mean?
Line 77-82: The objective of the paper is not really scientific. What are the scientific questions? What are the research hypotheses?
Material and Methods:
Line 92-93: flow or time proportional auto-sampling. Which sampling mode for which WWTP? Should be described.
Line 93: “temporal comparability was assumed for all samples”. Why?, What makes you make this assumption? Please explain.
Line 100: what is the ulrafiltration protocol used (type of membrane, filter, porosity, cut-off, ultrafiltration parameters...). Should be clearly described.
Line 105: Performance of each RT-qPCR assays (sensitivity, efficacity, specificity) used in the study should be described. Also data on RT-qPCR assays in supplementary files must be included in the main manuscript, as well as the reference for the primers and probes.
Line 111: Why is an average of the concentrations obtained by the three RT-PCR assays performed? This does not make sense. The three assays target different areas of the gene, which are most certainly degraded differently by environmental factors. Also, the RT-PCR efficiencies are certainly different (data not available) and prevent the calculation of such an average. This point should be explained and scientifically justified.
Line 115: details on the normalization are not clear. Please clarify.
Line 146-147: wastewater concentration explained in percentage? So strange, and finally none results are presented in this format in the figures. I don’t understand.
Line 151: details on laboratory comparison provided in Appendix D. I don’t understand how the figure provided allows for a comparison between laboratories. Please clarify.
Line 151-157: What is the rationale for the definition of the three proposed alternative indicators? How are they defined? According to which criteria?
Line 164: Why did the sampling occurred always at the same days of the week? This doesn't make sense to me, because we know that the composition of wastewater varies during the day but also during the days of the week, especially on weekends, with people preferring to be at home than at work. this variability is not taken into account at all in the study. Please explain and discuss this point.
Line 165-170: what is the interest of this part for the study? These are practical aspects of the monitoring set up, which are not relevant for the understanding of the results in my opinion. Please remove. However, it would be much more interesting and relevant to describe how the wastewater data was aggregated at national level (figure 1). There is no mention of this in the article, and the figures only show national results. This point must be explained and described in details.
Line 176: I don’t understand the equation for the estimation of the COVID-19 cases at the catchment area. Why is not not simply the sum of the case of each municipality contributed to the WWTP?
Line 183-184, please specify the version of the software used.
Results
Table 1: Personally, and in comparison, with other studies, I think that the correlation coefficients could be higher. I am not convinced that the right data are being used, especially regarding the number of cases. To me it does not make sense to use the number of new positive cases detected on the day of sampling. This does not take into account the incubation period before symptoms are reported and a positive test is obtained, nor does it take into account the period of excretion of infected persons. The number of genome copies detected in a given sample is not a result of new cases. New analyses should be envisaged on this point.
Also, why there are not determined values in the table?
Table 2: This table can be removed. All the information is described in the text. Moreover, no numeric data is provided on the impact of each uncertainty sources.
Discussion:
Line 265: Please explain what the role or mechanism of the different factors is cited.
The interest or not of normalization of the data (rainfall or fecal ratio) should be better discussed. Is it interesting or not to do that?
Reviewer 2 Report
Wastewater surveillance is now a recognised tool in supporting public health decisions in the management of the COVID-19 pandemic. The paper presents the outcomes of the Belgian national wastewater surveillance system for SARS-CoV-2, representing 45% of the population. Though data for such systems have been published for many countries, comparison of results through different outbreak waves and the use of data for public health measures is still of interest. However, further detail should be given on the methods and data interpretation for clarification.
More specifically:
Introduction:
- lines 66-67: crAssphage should also be mentioned as a potential faecal indicator for normalisation of wastewater data
Methods:
- line 93: What does this sentence mean? " Temporal comparability was assured for all samples."
- lines 99-100: What was the ultrafiltration method used by the different laboratories?
- Was there an interlaboratory comparison carried out between the laboratories? Were there any consistent differences identified?
- lines 116-117: Was the limit of quantification the same for all laboratories? How was it calculated?
- lines 142-163: This section describes the indicators derived from the measured concentrations in sewage. The section refers to 7 days moving average and 6 days increase. However, samples were collected twice a week (according to the next paragraph, on Mondays and Wednesdays, though it is not stated if this was the case for all WWTPs). Please explain how 6 days increase was calculated from this data.
- This section is unclear: "The estimated impacts of the uncertainties encountered on the three indicators are listed and discussed based on the work of Bertels et al. [13]. For visual interpretation, the arbitrarily estimated impacts are graded from over- (+), neutral- (+/-), and under-estimation (-).
- lines 164-171: The last paragraph is more suited to results or discussion section than to methods.
- lines 187-189: The last sentence of the paragraph is unclear. "The correlations were computed on raw data non-averaged, and several outbreak periods were selected visually from Figure 1."
Results:
- Section 3.1. Most studies investigate the effect of shifting data on the correlation between sewage data and case numbers. Was it tested on this dataset? With what result? The three curves on Figure 1 are very different, more explanation should be provided. Figure 1 is bad quality.
- Section 3.2. The presented indicators are useful for visualisation and public information purposes, but for public health decision makers the added value over sewage data is unclear and should be presented in more detail.
- lines 245-248. Was the discrepancy between viral concentration and viral load analysed statistically?
Discussion:
- lines 261-267: Some of the listed factors, e.g. vaccination should lead to lower rather than higher case numbers. This should be explained further.
General remarks: language should be checked throughout the manuscript. "Wastewater concentration" phrase is not ideal. Concentration in wastewater or wastewater data would be better.
Reviewer 3 Report
The manuscript presented by Janssens and colleagues is well written and present the authors' predictive model of SARS-CoV-2 waves based on analysis of wastewater in Belgium. The concept and their hypothesis are explained thoroughly and I only have minor comments.
- Figure 1: We see an inflection of all 3 markers between August and October 2021. However no wave was observed. How are these coordinated increases of the 3 risk factors explained within your model?
- Such models and observations were generated in the context limited population movements (whether it was enforced or simply due to the general fear of traveling). How does the global reopening of national and international travel will impact such models. Authors may want to add a few sentences within the discussion.